# Brief exposure to social media during the COVID-19 pandemic: Doom-scrolling has negative emotional consequences, but kindness-scrolling does not

Kathryn Buchanan[1]*, Lara B. Aknin[2], Shaaba Lotun[1], Gillian M. Sandstrom[1]

**1** Department of Psychology, University of Essex, Essex, United Kingdom, **2** Department of Psychology, Simon Fraser University, Burnaby, British Columbia, Canada

* k.buchanan@essex.ac.uk

**Data Availability Statement:** The data for the Twitter study (DOI: 10.17605/OSF.IO/5YA9M) and the YouTube study (DOI: 10.17605/OSF.IO/A584S) are available on the OSF repository.

## Abstract

People often seek out information as a means of coping with challenging situations. Attuning to negative information can be adaptive because it alerts people to the risks in their environment, thereby preparing them for similar threats in the future. But is this behaviour adaptive during a pandemic when bad news is ubiquitous? We examine the emotional consequences of exposure to brief snippets of COVID-related news via a Twitter feed (Study 1), or a YouTube reaction video (Study 2). Compared to a no-information exposure group, consumption of just 2–4 minutes of COVID-related news led to immediate and significant reductions in positive affect (Studies 1 and 2) and optimism (Study 2). Exposure to COVID-related kind acts did not have the same negative consequences, suggesting that not all social media exposure is detrimental for well-being. We discuss strategies to counteract the negative emotional consequences of exposure to negative news on social media.

## Introduction

Stories about COVID-19 have dominated the news and have been shared widely on social media since early 2020, when COVID was labelled a pandemic by the World Health Organization. On any given day, people are exposed to multiple stories about: government regulations and lifestyle restrictions; people protesting such restrictions and breaking rules; scientific discoveries (e.g., treatments, vaccines, variants); and shortages in supply chain issues (e.g., PPE, hospital beds, vaccines). Over the same timeframe, average levels of anxiety and depression have increased [1–3] and people in various countries are reporting increased symptoms of serious psychological distress [4, 5]. How can people cope?

One way that people typically handle tumultuous times and uncertainty is by seeking information. Various models of threat perception [6, 7] suggest that negative emotions lead people to "reassert control over the situation" [8, p192] by seeking out knowledge about a novel context or threat. For instance, someone experiencing fear about an upcoming medical exam may search for, or be receptive to, information about exam procedures, typical timelines, and

**Funding:** The authors received no specific funding for this work.

**Competing interests:** The authors have declared that no competing interests exist.

outcomes as a coping strategy. This desire may help explain why people are drawn to negative news stories [9], which offer facts about potential threats and advice on how to manage them. In short, gathering information helps people make sense of their environment, and in turn, can improve their well-being.

While this information gathering strategy may prove advantageous in most situations, the extraordinary nature of the pandemic provides an extreme test of its utility. In particular, the pandemic presents a significant upheaval and challenge for mental health because it is a long-term, widespread, overwhelming, and multi-dimensional threat [10]. Indeed, information-seeking during the pandemic may prove problematic because negative information is ubiquitous and unending, and no amount of information can eliminate the pervasive sense of uncertainty [11]. Thus, even if people have successfully employed this strategy in other situations before, they may find that seeking information amplifies their concerns and undermines their well-being during the pandemic. Of course, people need to remain at least minimally informed about fast-evolving public health measures implemented to protect society, which means that most people are exposed to some COVID-related news.

How much COVID-related content are people consuming though, and how does it impact their well-being? Early evidence suggests that many people are spending time every day consuming COVID-related content—sometimes called *doom-scrolling* when one becomes caught in an unending cycle of negative news—and that doing so is associated with poorer mental health. For instance, data from a large cross-sectional survey conducted with over 69,000 college students in France between April-May 2020, when many countries were imposing lockdowns and stay-at-home orders, found that 45% of respondents spent more than half an hour consuming pandemic-related informational content each day [12]. Moreover, individuals who reported spending more time consulting COVID-related news each day also reported higher levels of anxiety, distress, stress, and depression. These findings align with a large cross-sectional study of over 6,300 Americans conducted in March 2020 which found that people spent on average 55 minutes each day on social media and were seeking COVID-19 related information through various traditional news sources as well [13]. Time spent on social media, and the number of news sources consulted both independently predicted greater mental distress, even when controlling for demographics, previous depressive symptoms or psychiatric illness, and perceived risk of catching the coronavirus. Similarly, [14] found that frequent social media use during the pandemic was related to negative mental health effects. Other studies have used longitudinal designs to examine within-person variation in time use and well-being and have yielded similar results. For instance, two panel studies tracking time use during the early months of lockdown in the UK [15] and Ireland [16] found that time spent procuring news about COVID was one of the least enjoyable activities in a day, again suggesting that seeking out COVID-related news may undermine positive emotions. Taken together, the existing data suggest that extended exposure to COVID information is linked with lower well-being.

Exposure to information about COVID may prove harmful for one's well-being, but the current evidence is unclear for several reasons. First, existing findings are limited by the correlational nature of the data, which cannot rule out the possibility of reverse causality: people with poorer mental health, or who are feeling unhappy and anxious, may seek out more negative news about the pandemic. To determine whether COVID news has a negative causal effect on well-being, participants need to be randomly assigned to either consume COVID news or not. Second, most research has asked people to estimate the amount of time they spend consuming COVID-related news on average each day [12, 13], or how much time they spent on a particular day [15]. While self-report estimates are often necessary in large questionnaires, this methodology may provide inaccurate estimates of the actual amount of time spent consuming COVID-related news, and in turn, its impact on well-being. An experimental design with

control over exposure time could address this concern. Finally, while past work suggests that spending hours of time reading COVID-related news is negatively associated with well-being, little is known about whether brief exposure to COVID-related news is also detrimental.

Accordingly, we tested whether 2–4 minutes of COVID-related news could have aversive emotional impacts relative to a no-information control condition. In minimizing the duration of exposure to just a few minutes, we provide a conservative test of just how powerful an effect COVID-related news may have on people's emotions. If even a mere few minutes of exposure to COVID-related news can result in immediate reductions to well-being, then extended and repeated exposure may over time add up to significant mental health consequences. Indeed, such a theorisation is in line with acknowledgements that even so called "small effects" obtained in experiments can have the capacity for real world consequences, particularly in situations where effects may accumulate over time [17, 18]. Moreover, our work also helps to address the urgent need for research on the impact of media consumption around COVID-19 on mental health [19].

### Current research

We examined the emotional consequences of brief exposure to authentic COVID-related information on two social media platforms: Twitter (Study 1) and YouTube (Study 2). In light of past research documenting a negative association between COVID information-seeking and well-being [12, 15], we pre-registered the hypothesis that even minimal exposure to COVID information would have negative emotional consequences, compared to a no information control. However, not all information about COVID is negative. The COVID pandemic has encouraged novel acts of kindness, both small and large [20], which are often documented and shared in news and social media [21]. We hypothesized that exposure to the acts of kindness elicited by COVID would not have the same negative emotional consequences.

### Transparency statement

We originally hypothesized that reading about COVID-related acts of kindness would not only avoid the adverse impacts of exposure to COVID information, but actually have a positive effect on mood compared to the no information control condition. This hypothesis was not supported, perhaps because it is not possible for people to separate the positivity of kind acts from the negative context that inspired them. In order to test that hypothesis, we included an additional "amusement" condition, which was intended to rule out the explanation that the positive effects of the kindness condition were simply due to it increasing positive affect. For reasons of parsimony, and to avoid confusion, we do not report the responses from the amusement condition, or the responses on measures that are germane to this unsupported hypothesis (that the kindness condition would have more positive outcomes than the no information control condition, e.g., elevation). For more information, see the data and materials, which are posted online.

## Study 1: Twitter

The pre-registration, materials, data and analysis script for this study can be found on the OSF (https://osf.io/5ya9m/?view_only=13a1cf936373404698d94d3a0963ed2b).

### Method

**Ethics approval.**   Prior to commencement of this research, both Studies 1 and 2 were reviewed by the Faculty of Science and Health Ethics Committee at the University of Essex

and granted approval with the following code: ETH1920-1274. In both Studies 1 and 2, written consent was obtained from participants prior to their participation in the online studies.

**Participants.** An a priori power analysis in G*Power indicated that a sample of 500 participants (125 in each of our four original conditions) was required to detect a small-to-medium effect (f = .15) with 80% power and a one-tailed alpha of .05. A total of 665 people started the survey, and due to experimenter error applying exclusion criteria, we thought we had surpassed our recruitment target. However, a total of 402 participants consented, were at least 18 years of age, and completed the survey before our pre-registered cut-off date, meaning that we did not reach our recruitment target. We report the findings from the participants ($N = 299$) who completed the study in the three conditions relevant to our analyses ($M_{age} = 29$, SD = 12; 74% female, 17% male, 9% identified in another way; 86% White, 1% Black, 3% Asian, 6% Mixed, 4% Other); we do not report the results from a fourth ("amusement") condition (see Transparency Statement above).

Most participants (76%) had taken part in a previous, unrelated study in our lab, and had expressed an interest in participating in future studies; we recruited them via email. We also recruited via university-wide mailing lists, the departmental subject pool, and personal social media accounts.

**Procedure.** Participants were randomly assigned to one of three conditions: COVID-information, COVID-kindness, or no information control. Participants in the first two conditions were shown the live contents of a Twitter feed for at least 2 minutes (controlled by a timer on Qualtrics); participants in the no information control condition were not shown a Twitter feed and were simply directed to the well-being measures below. To increase ecological validity, participants in the COVID-information and the COVID-kindness conditions viewed the contents of real-time Twitter feeds (@PandemicCovid20 and @covidkind, respectively), meaning that different participants were exposed to different tweets. Specifically, the COVID-information tweets included topics such as escalating COVID figures, long-term health implications of COIVD, and laboratory results indicating abnormalities in pregnant women. In contrast, examples of tweets in the COVID-kindness condition were "My daughter made me a fancy bow tie to wear for [online] meetings and classes", "This 99-year-old great-grandmother has a lot to celebrate after recovering from COVID-19" and "When you have long [online] meetings and your daughter sets up a refreshment trolley". After the manipulation, participants reported their current positive and negative affect and their current optimism.

**Measures.** *Positive and negative affect*. The Scale of Positive and Negative Experience [22] consists of 12 items that describe feelings which reflect two subscales: positive affect (e.g., "*pleasant*") and negative affect (e.g., "*unpleasant*"). Participants indicated the extent to which they were experiencing each feeling "*right now*" using a 5-point scale (1 = *Very slightly or Not at all*, 5 = *Extremely*). The reliability of the positive affect scale was excellent ($\alpha = .91$), and the reliability of the negative affect scale was good ($\alpha = .85$).

*State optimism*. The State Optimism Measure [23] consists of 7 items that measures people's tendency to feel positive about the future (e.g., "*Right now, I expect things to work out for the best*"). Participants responded to each item in reference to how they were feeling "right now" using a 5-point scale (1 = *Strongly disagree*, 5 = *Strongly agree*). The reliability of the SOM was excellent ($\alpha = .91$).

## Results

All analyses were pre-registered. We conducted a series of ANOVAs, followed by pairwise comparisons, to examine whether exposure to COVID information impacted current emotion and optimism. The ANOVAs revealed a significant effect of condition on positive affect, $F(2,$

**Table 1. Pairwise comparisons between the no information control condition and each of the other conditions in Studies 1 (Twitter) and 2 (YouTube).**

| | M (SD) | | | COVID-information vs. no information control | | | | COVID-kindness vs. no information control | | | |
|---|---|---|---|---|---|---|---|---|---|---|---|
| | COVID-information | no information control | COVID- kindness | t | p | 95% CI | d | t | p | 95% CI | d |
| Study 1 (Twitter) | | | | | | | | | | | |
| Positive affect | 2.38 (0.85) | 2.74 (0.90) | 2.70 (0.84) | **-2.92** | **0.01** | **[-0.60, -0.12]** | **0.42** | -0.33 | 0.99 | [-0.28, 0.20] | 0.05 |
| Negative affect | 2.20 (0.81) | 2.01 (0.83) | 2.12 (0.82) | 1.69 | 0.28 | [-0.03, 0.42] | 0.24 | 0.99 | 0.97 | [-0.11, 0.34] | 0.14 |
| Optimism | 2.98 (0.98) | 3.08 (0.91) | 3.00 (0.93) | -0.77 | 0.99 | [-0.37, 0.16] | 0.11 | -0.65 | 0.99 | [-0.34, 0.17] | 0.09 |
| Study 2 (YouTube) | | | | | | | | | | | |
| Positive affect | 2.26 (0.80) | 2.59 (0.84) | 2.78 (0.93) | **-4.05** | **< .001** | **[-0.49, -0.17]** | **0.40** | 2.18 | 0.09 | [0.02, 0.36] | 0.22 |
| Negative affect | 2.37 (0.90) | 2.18 (0.89) | 1.88 (0.81) | 2.11 | 0.11 | [0.01, 0.36] | 0.21 | **-3.62** | **0.001** | **[-0.47, -0.14]** | **0.36** |
| Optimism | 2.77 (0.83) | 3.02 (0.93) | 3.09 (0.88) | **-2.84** | **0.01** | **[-0.42, -0.08]** | **0.28** | 0.75 | 0.99 | [-0.11, 0.24] | 0.07 |

NOTE: p-values have been multiplied by three to reflect a Bonferroni correction, and significant effects are in bold. Effect sizes computed following Lakens [24]).

296) = 5.18, $p = .01$, $\eta_p^2 = .03$, but not negative affect, $F(2, 296) = 1.45$, $p = .24$, $\eta_p^2 = .01$, or optimism, $F(2, 296) = 0.35$, $p = .71$, $\eta_p^2 = .002$. Follow-up analyses, using Bonferroni corrections to adjust for multiple comparisons, revealed that participants in the COVID-information (i.e., doom-scrolling) condition reported lower levels of positive affect than participants in the no information control group (see Table 1 and Fig 1). Levels of negative affect and optimism did

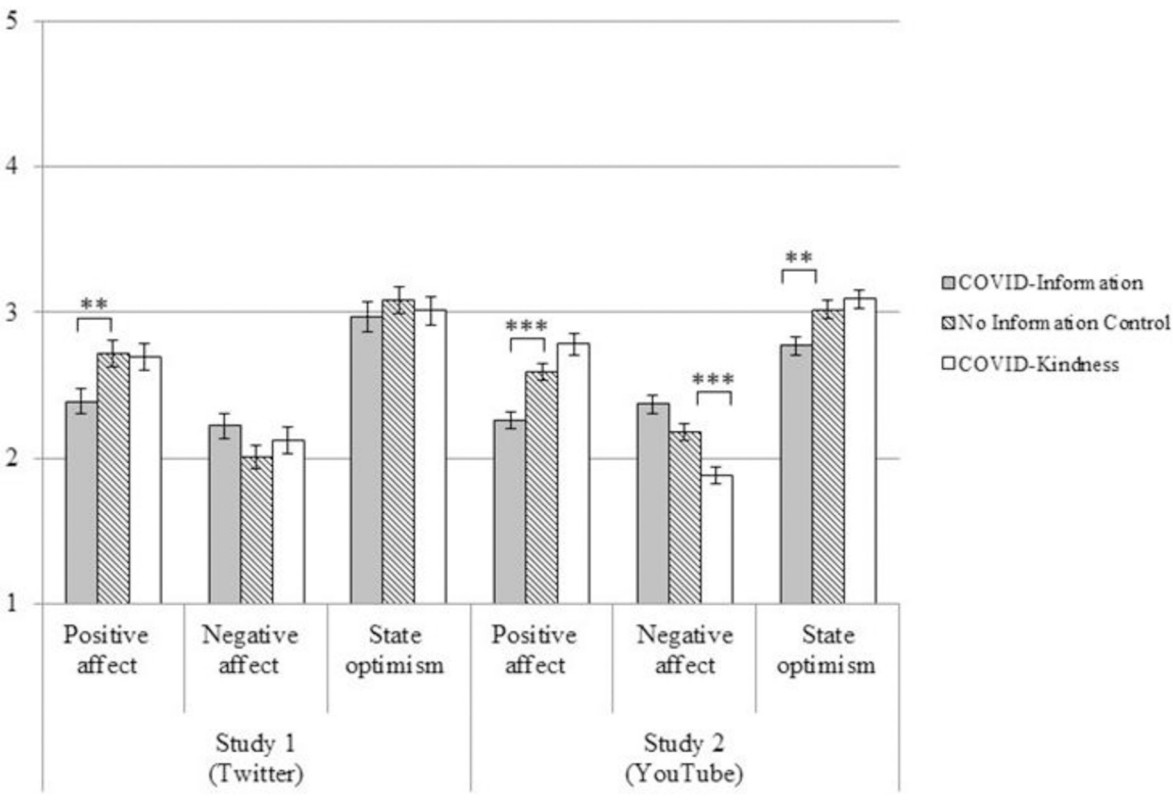

**Fig 1. Impact of condition on well-being for Studies 1 and 2.** Shown are the results from post-hoc pairwise t-tests depicting well-being differences between the no information control condition and the COVID-kindness and COVID-information conditions. Error bars represent standard errors. ** $p < .01$, *** $p < .001$.

not differ between the COVID-information and no information control conditions (see Table 1 and Fig 1).

Next, we tested whether reading about COVID-related acts of kindness produced the same negative mood consequences as reading about COVID news in general. It did not; participants in the COVID-kindness condition reported similar levels of positive affect, negative affect, and optimism compared to people in the no information condition (see Table 1 and Fig 1).

In other words, briefly reading about COVID-related news reduced positive affect but did not increase feelings of negative affect nor undermine feelings of optimism. However, not all information about COVID had negative consequences; there was no evidence to suggest that reading about COVID-related acts of kindness has the same negative effects.

## Study 2: YouTube

In Study 2, we examined whether minimal exposure to COVID information would also have detrimental well-being effects on a social media platform that uses a different primary form of communication: video. We recruited a YouTube creator (a performer who regularly uploads videos of their lives or opinions to a mass online audience) to create the stimuli. This person has a channel on YouTube, and posts video content that is largely viewed by a regular community of more than 600k subscribers but can be viewed by anyone. We recruited viewers who were familiar with this creator as our participants, reasoning that this parasocial relationship (i.e., a one-sided relationship, such as a person has with a favourite television personality or fictional character; [25]) might amplify the mood effects of consuming social media, through emotional contagion [26]. Parasocial relationships can compel our purchasing decisions [27], and influence the way we vote in presidential elections [28]. Further, people feel emotionally connected to their parasocial ties, experiencing grief when parasocial relationships end [29]. It therefore seems reasonable to expect that people will be responsive to the cues and emotions of known parasocial others, such as YouTube creators.

The pre-registration, materials, data and analysis script for this study can be found on the OSF (https://osf.io/a584s/?view_only=bbeb53ac15124a3786b117f5590c1b48).

### Method

**Participants.**   The same power analysis as in Study 1 suggested that we again recruit 500 participants (125 in each of our four original conditions). As in Study 1, we allowed data collection to continue for a full week, which resulted in us exceeding our target sample size. A total of 1642 people started the survey, and 813 participants consented, completed the survey, were at least 18 years of age, and passed pre-registered attention checks. As in Study 1, here we report the findings from the participants ($N$ = 603) who completed the study in the three conditions relevant to our analyses ($M$age = 26, $SD$ = 9; 59% female, 23% male, 18% identified in another way; 84% White, 1% Black, 4% Asian, 6% Mixed, 4% Other, 1% did not report their ethnicity).

The YouTube creator who featured in the stimuli, and a second YouTube creator with an overlap in audience, both recruited participants by promoting the study on their individual YouTube channels.

**Procedure.**   Participants were randomly assigned to one of three conditions: COVID-information, COVID-kindness, or no information control. Participants in the first two conditions watched a reaction video, in which a YouTube creator reacted to either news about COVID (e.g., doctors protesting the lack of personal protective equipment), or COVID-related acts of kindness (e.g., leaving thank you gifts for a delivery driver) for 4 to 4.5 minutes (see OSF for stimuli); participants in the no information control group did not watch a video.

**Measures.**   Participants completed the same positive affect, negative affect, and optimism measures as in Study 1. The measures were found to have good-excellent reliability (positive affect: $\alpha = .91$; negative affect: $\alpha = .87$; optimism: $\alpha = .89$).

## Results

All analyses were pre-registered. As in Study 1, we conducted a series of ANOVAs, followed by pairwise comparisons, to examine whether exposure to COVID information impacted current emotion and optimism. The ANOVAs revealed significant differences between conditions in positive affect, $F(2, 600) = 18.56$, $p < .001$, $\eta_p^2 = .06$, negative affect, $F(2, 600) = 16.19$, $p < .001$, $\eta_p^2 = .05$, and optimism, $F(2, 600) = 7.00$, $p = .001$, $\eta_p^2 = .02$. Follow up analyses with Bonferroni corrections revealed that, as in Study 1, participants in the COVID-information condition reported less positive affect than people in the no information control condition, but no difference in negative affect (see Table 1 and Fig 1). Unlike in Study 1, participants in the COVID-information condition also reported less optimism than people in the no information control condition.

Next, we tested whether participants who read about COVID-related acts of kindness experienced the same negative mood consequences as reading about COVID news in general. As in Study 1, participants in the COVID-kindness condition reported no difference in positive affect, or optimism compared to people in the no information condition (see Table 1 and Fig 1). Not only did participants in the COVID-kindness condition report no negative mood consequences on the positive affect and optimism measures, but they actually reported lower negative affect, compared to people in the no information condition.

To summarize, hearing news about COVID in a YouTube reaction video negatively affected people's mood in the same way as did reading similar information in a Twitter feed. However, hearing about COVID-related acts of kindness (i.e., kind-scrolling) not only avoided the negative effects of doom-scrolling, but lifted mood compared to the control condition.

## General discussion

The uncertainty of the COVID-19 pandemic has been associated with a stark increase in mental health concerns [11] and social media use. While existing evidence from correlational and longitudinal studies [12–16] suggests that these two occurrences may be related—specifically that increased exposure to COVID-related news may undermine well-being—the present studies offer the first experimental test of this possibility. In two studies, people randomly assigned to spend a few minutes consuming COVID-related information, either by reading a real-time Twitter feed (Study 1) or watching a YouTube reaction video (Study 2), reported lower well-being compared to a no treatment control group. Critically, the negative effects of COVID-information emerged with as little as two minutes of exposure. Consuming stories about COVID-related acts of kindness did not have the same negative consequences, and indeed in Study 2, reduced negative affect compared to the no treatment control group.

This work offers several new insights. First, it extends previous research by demonstrating a causal effect of exposure to negative news on emotional outcomes. Existing research on social media exposure to COVID-related news has been correlational, leaving open the possibility that unhappy people are more likely to seek out negative news. Second, the current research demonstrates that as little as *two minutes* of exposure to negative news about COVID-19 can have negative consequences. Given that many people spend more than 5–10 times the amount of time interacting with COVID-related news each day than our brief manipulation required (e.g., [12] find that 45% of their sample spent 30-minutes consuming COVID-related media daily), the present findings likely offer a conservative estimate of the emotional toll of interacting with

COVID-related media in the real world. Third, by demonstrating that exposure to COVID-related acts of kindness does not produce the same negative outcomes (and indeed may produce positive outcomes), we show that it is not simply that time spent on social media is problematic, but rather that consumption of *negative news* is the source of concern. This points the way to interventions that could limit the negative consequences of social media consumption.

## Implications

Our findings suggest the importance of being mindful of one's own news consumption, especially on social media. In some countries, news consumption via social media is on the rise, even though people acknowledge that news on these platforms has lower quality, accuracy, trustworthiness and impartiality; half of adults in the U.K. now use social media to keep up with the news, including 16% who use Twitter, and 35% who use Facebook [30]. People seek out social media for many reasons other than news consumption and may not realize that minimal exposure to negative news on these platforms can have such negative consequences.

If even a few minutes of exposure to COVID news on social media can have negative emotional consequences, what can be done? One strategy might be to limit one's exposure, but this may be difficult because people need to seek information, if only to comply with ever-changing government regulations and recommendations. Armed with this knowledge, government agencies could be mindful that the human need for information during times of stress comes with negative consequences, and they could proactively offer key information and guidelines in a brief and digestible manner. Doing so may satiate citizens' hunger for facts and offer clear guidance on how to respond. Critically, brief updates may minimize psychological costs that are associated with heavy consumption and dissuade people from searching for information elsewhere.

One strategy that individuals could employ would be to attempt to undo the negative by balancing it out with positive information [31]. Such an approach would be consistent with recent calls for traditional news media to report one positive story for every three negative stories [32]. Indeed, the algorithms that select the messages we are exposed to on social media could be modified to take valence into account and prioritize exposure for user well-being, instead of endless engagement. While such changes may be hard to imagine at the structural level, our findings highlight how social influencers can use their platforms to benefit their followers' well-being, through creating content that features kindness. There seems to be an appetite for positive content; in response to COVID, actor John Krasinski started a YouTube channel called "Some Good News", which now has 2.5 million subscribers. These efforts may be especially important in times of crisis, such as a global epidemic, when negative news is nearly impossible to avoid. We found that reading about COVID-related acts of kindness did not produce the same negative consequences as reading COVID news, and that hearing about acts of kindness from a known creator on YouTube produced less negative affect than no exposure at all, suggesting that people might want to balance out the doom-scrolling with kindness-scrolling.

Finally, another possible "undoing" strategy is to take matters into one's own hands and engage in active coping after exposure to COVID news. People may be able to maintain their well-being during stressful times by investing time in activities such as helping others [16, 33, 34] and meeting their own basic psychological needs [35]. Recent research suggests that these activities can help to bolster happiness, even during the pandemic.

## Limitations and future directions

We presented COVID-19 news via two social media platforms and found generally consistent results: consumption of a mere 2–4 minutes of COVID-related news led to immediate and

significant reductions in positive affect. Given that exposure to COVID-related acts of kindness on the same platform did not result in the same negative consequences, the evidence suggests that it is negative news itself that causes problems, rather than any specific platform; there is reason to expect that consumption of negative news from traditional news sources would yield similar outcomes. Social media may, however, be especially problematic; although one needs to actively seek out news via traditional media (e.g., newspapers, websites), the nature of platforms like Twitter and Facebook make passive consumption of news almost unavoidable, even if one ventures to these sites strictly for social purposes. We did, however, find differences between the two social media platforms we examined, and more work is needed to understand the mechanisms that drive these differences. This may be partly explained by the fact that watching a video is more engaging than reading a tweet. Additionally, in our YouTube study participants were exposed not only to the news stimuli, but also the YouTube creator's *reaction* to these stimuli. Indeed, the creator's reaction may offer guidance on how to interpret the information, especially if one already has a parasocial relationship with that person, as was the case with our participants.

One strength of our Twitter study is that we utilised real Twitter feeds that continuously updated with new content, providing a naturalistic design that increases the external validity of our results. However, this design also means that we had no control over the content, and each participant may have experienced slightly different stimuli, even compared to other participants in the same condition. We do not know how objectively negative the COVID-information feed was, or how objectively positive the COVID-kindness feed was, and more work is needed to understand what features of content affect participants' emotional responses.

Notably while our research was not limited to samples comprised of undergraduate psychology students or MTurk participants, the current research is limited by its use of two relatively homogenous samples, which featured primarily young white and female participants, thereby narrowing the generalisability of the results (Study 1: $M_{age}$ = 29, SD = 12; 74% white; Study 2: $M$age = 26, $SD$ = 9; 59% female). Indeed, some research has found that younger females have fared worse than their male counterparts in terms of mental health and stress during the pandemic [14, 36]. On the other hand, [37] found males experienced larger reductions in positive affect than females during the pandemic, and [14] found frequent social media use during the pandemic was associated with negative mental health consequences for both males and females. Taken together, it is unclear what effect a more balanced gender sample may have had on the results. With respect to homogeneity of age, given that older adults are able to regulate emotions more effectively than younger adults [38, 39], we might speculate that an older sample may not have been as negatively impacted by brief exposure to COVID-info on social media. However, older adults may also be less likely to use social media [40], which reduces the importance of this limitation.

More research is needed to test the generalizability of our findings, and to explore possible mediators. For instance, future work could examine whether exposure to brief information about other large-scale and all-encompassing threats, such as climate change, may lead to similar emotional consequences. However, it is possible that immediacy acts as a mediator; despite the efforts of a wide range of scientists and private citizens to educate the public about climate change, it remains psychologically distant to many.

## Conclusion

Although information-seeking is generally an adaptive coping strategy in times of threat, doing so during a pandemic may be less helpful. Unlike most world events, the threat of the current pandemic affects many life domains (relationships, education, work, leisure), and

there is uncertainty about how long it will last, and what will happen next. Even a few minutes of exposure to COVID-related news on social media can ruin a person's mood. We would all do well to be mindful of these effects and consider balancing our doom-scrolling with some kindness-scrolling.

## Acknowledgments

The authors thank Paulo Ferreira for technical assistance in Study 1, and Jamie Raines for recording the stimuli in Study 2. They also thank ESSEXLAB for seedcorn funding that helped build the participant pool that was used in Study 1.

## Author Contributions

**Conceptualization:** Kathryn Buchanan, Lara B. Aknin, Gillian M. Sandstrom.

**Data curation:** Kathryn Buchanan, Shaaba Lotun, Gillian M. Sandstrom.

**Formal analysis:** Kathryn Buchanan, Shaaba Lotun, Gillian M. Sandstrom.

**Investigation:** Kathryn Buchanan, Lara B. Aknin, Shaaba Lotun, Gillian M. Sandstrom.

**Methodology:** Kathryn Buchanan, Lara B. Aknin, Shaaba Lotun, Gillian M. Sandstrom.

**Project administration:** Lara B. Aknin, Gillian M. Sandstrom.

**Resources:** Kathryn Buchanan, Lara B. Aknin, Shaaba Lotun, Gillian M. Sandstrom.

**Software:** Gillian M. Sandstrom.

**Supervision:** Lara B. Aknin, Gillian M. Sandstrom.

**Validation:** Shaaba Lotun, Gillian M. Sandstrom.

**Visualization:** Shaaba Lotun.

**Writing – original draft:** Lara B. Aknin, Gillian M. Sandstrom.

**Writing – review & editing:** Kathryn Buchanan, Lara B. Aknin, Shaaba Lotun, Gillian M. Sandstrom.

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
