## [Decision Letter · Decision Letter 0]

2 Aug 2021

PONE-D-21-14745

Exposure to social media during the COVID-19 pandemic: Doom-scrolling has negative emotional consequences, but kindness-scrolling does not

PLOS ONE

Dear Dr. Buchanan,

Thank you for submitting your manuscript to PLOS ONE. After careful consideration, we feel that it has merit but does not fully meet PLOS ONE’s publication criteria as it currently stands. Therefore, we invite you to submit a revised version of the manuscript that addresses the points raised during the review process.

The paper needs a MINOR REVISION in order to be considered for a publication. Please, follow the suggestion given by the reviewers in order to improve the quality of the paper.

We look forward to receiving your revised manuscript.

Kind regards,

Barbara Guidi

Academic Editor

PLOS ONE

2. Please note that in order to use the direct billing option the corresponding author must be affiliated with the chosen institute. Please either amend your manuscript to change the affiliation or corresponding author, or email us at plosone@plos.org with a request to remove this option.

4. Please provide additional details regarding participant consent. In the ethics statement in the Methods and online submission information, please ensure that you have specified whether consent was informed.

5.  Please include a caption for figure 2.

Additional Editor Comments (if provided):

Reviewers' comments:

Reviewer's Responses to Questions

**Comments to the Author**

1. Is the manuscript technically sound, and do the data support the conclusions?

Reviewer #1: Yes

Reviewer #2: Yes

2. Has the statistical analysis been performed appropriately and rigorously? 

Reviewer #1: Yes

Reviewer #2: Yes

3. Have the authors made all data underlying the findings in their manuscript fully available?

Reviewer #1: Yes

Reviewer #2: Yes

4. Is the manuscript presented in an intelligible fashion and written in standard English?

Reviewer #1: Yes

Reviewer #2: Yes

5. Review Comments to the Author

Reviewer #1: Review article: Exposure to social media during the COVID-19 pandemic: Doom-scrolling has negative emotional consequences, but kindness-scrolling does not

In this paper, the authors examine the emotional consequences of exposure to Covid-related news via Twitter and YouTube videos. In particular, they aim to verify if a brief engagement (such as 2-4 minutes) could affect the well-being of the users.

To achieve this result, they perform two studies: one to investigate the impact of written news (Twitter) and the second one to analyze the influence of videos (YouTube). They consider three conditions: the impact of news about Covid (Covid-information) and positive information (Covid-kindness), and no information control. And, as usual, the participants were randomly assigned to one of the three conditions.

They show that even very brief exposure to Covid news can have negative consequences on mental healths.

Evaluation:

The mental health of users during the Covid pandemic is a remarkable subject, and the authors report their analysis about the topic in detail. The background, the results and the used methods are well-written and motivated. Moreover, the chosen approach (to use two different social platforms) helps to deeper analyze the complex effects of social networks on users' behavior.

I appreciate that the authors provide an interesting comparison with the current state of the art. Nevertheless, I suggest improving the clarification about the "brief" exposure. It is not clear (above all in the introduction) why to use the short time of 2-4 minutes. Even if the result is interesting, adding stronger motivation could increase the article's relevance.

In my opinion, the paper is ready for publication.

Reviewer #2: This is a compelling and timely article on the emotional impact of exposure to information during the COVID-19. I found the writing and methods clear and table and figures useful for communicating the main findings. Below are a few suggestions for improvement:

-I found the title's reference to doom scrolling a bit misleading given the methods included exposure to very brief exposures to twitter and you tube

-I'm curious if the authors have a sense of why there was such a large gender imbalance in the first study - only 17% male seems unexpected

- It would be helpful if the authors could provide more details about the measures used in the study (e.g. sample items, more on psychometrics etc.)

-I'm confused by why the authors said they weren't going to report the kindness data (amusement statement - I'm unfamiliar with this term) in the transparency statement, but then display it in Table 1.

-The limitations section does not really name methodological limitations of the current study (e.g. relatively homogenous sample). I'd suggest expanding this section.

6. PLOS authors have the option to publish the peer review history of their article (what does this mean?). If published, this will include your full peer review and any attached files.

Reviewer #1: No

Reviewer #2: **Yes: **Rebekka Lee

---

## [Author Response · Author response to Decision Letter 0]

18 Aug 2021

August 17, 2021

Dear Dr. Guidi,

My co-authors and I thank you and the reviewers for your helpful feedback on our manuscript, now entitled “Brief exposure to social media during the COVID-19 pandemic: Doom-scrolling has negative emotional consequences, but kindness-scrolling does not” (PONE-D-21-14745). We have responded to the queries and suggestions from the editorial team listed below, and are pleased to submit our manuscript for further consideration at PLOS ONE. 

RESPONSE: We have updated our manuscript to meet the PLOS ONE style template. 

2. Please note that in order to use the direct billing option the corresponding author must be affiliated with the chosen institute. Please either amend your manuscript to change the affiliation or corresponding author, or email us at plosone@plos.org with a request to remove this option.

RESPONSE: We have confirmed with the lead author’s institution that we have now provided the correct billing arrangements. We had erroneously selected the wrong option in our original submission and have now emailed the above address to request that this be changed. 

RESPONSE: As requested, we have added the following statement to the Method section of Study 1 under the subheading “Ethics Approval”. Please see page 7 of the revised manuscript.

“Prior to commencement of this research, both Studies 1 and 2 were reviewed by the Faculty of Science and Health Ethics Committee at the University of Essex and granted approval with the following code: ETH1920-1274.”

4. Please provide additional details regarding participant consent. In the ethics statement in the Methods and online submission information, please ensure that you have specified whether consent was informed.

RESPONSE: As requested, we now mention that participants provided informed consent in the revised manuscript. The information has been incorporated into the Method section of Study 1 under the subheading “Ethics Approval”. Please see page 7 of the revised manuscript.

“Prior to commencement…. In both Studies 1 and 2, written consent was obtained from participants prior to their participation in the online studies”

5. Please include a caption for figure 2.

RESPONSE: Our apologies for not including a figure caption in our original submission. We have included a caption for Figure 2 in our revised manuscript. 

RESPONSE: We could not find any instances of retracted articles in our reference list. However, I realised that I had cited an unpublished paper (Buchanan, 2021) that is not available on a pre-print server so have removed reference to this. We have also double checked our reference list to ensure all papers cited are in the reference list and vice versa. 

Reviewers' comments:

Reviewer #1: I appreciate that the authors provide an interesting comparison with the current state of the art. Nevertheless, I suggest improving the clarification about the "brief" exposure. It is not clear (above all in the introduction) why to use the short time of 2-4 minutes. Even if the result is interesting, adding stronger motivation could increase the article's relevance.

RESPONSE: We thank Reviewer 1 for encouraging us to justify our focus on brief COVID-19 media exposure and have taken the opportunity to do so in our revision. Specifically, on pages 5-6 of the revised manuscript we now provide our rationalization as follows:“…we tested whether 2-4 minutes of COVID-related news could have aversive emotional impacts relative to a no-information control condition. In minimizing the duration of exposure to just a few minutes, we provide a conservative test of just how powerful an effect COVID-related news may have on people’s emotions. If even a mere few minutes of exposure to COVID-related news can result in immediate reductions to well-being, then extended and repeated exposure may over time add up to significant mental health consequences. Indeed, such a theorisation is in line with acknowledgements that even so called “small effects” obtained in experiments can have the capacity for real world consequences, particularly in situations where effects may accumulate over time [16, 17]. Moreover, our work also helps to address the urgent need for research on the impact of media consumption around COVID-19 on mental health [18].” 

Reviewer #2: Below are a few suggestions for improvement:

a) I found the title's reference to doom scrolling a bit misleading given the methods included exposure to very brief exposures to twitter and you tube

RESPONSE: We agree that it is important to emphasize the brief nature of our manipulation and, as such, have amended the title of our paper to more appropriately reflect this critical detail. The revised title of the current manuscript is: “Brief exposure to social media during the COVID-19 pandemic: Doom-scrolling has negative emotional consequences, but kindness-scrolling does not.”

b) I'm curious if the authors have a sense of why there was such a large gender imbalance in the first study - only 17% male seems unexpected

RESPONSE: We agree that the gender ratio in Study 1 is unbalanced and can only speculate on why this may have occurred. One possibility is that our sample was skewed towards females in Study 1 because participants had taken part in a previous experiment and they had indicated an interest in participating in future studies. To the extent that volunteering to help in future studies is seen as a prosocial behaviour and females tend to be more prosocial than female this may reflect the bias in the sample size. However, there could be other factors at play. According to Smith (2008), the higher participation of women in online studies is not uncommon. 

Smith, G. (2008). Does gender influence online survey participation?: A record-linkage analysis of university faculty online survey response behavior. ERIC Document Reproduction Service No. ED 501717

c) It would be helpful if the authors could provide more details about the measures used in the study (e.g. sample items, more on psychometrics etc.)

RESPONSE: As requested we have included further details about the measures used. For example, on pages 8-9 and 13 we now provide sample items, scales, classification of the reliability ratings.

d) I'm confused by why the authors said they weren't going to report the kindness data (amusement statement - I'm unfamiliar with this term) in the transparency statement, but then display it in Table 1.

RESPONSE: We thank Reviewer 2 for pointing out this inconsistency and have now clarified the transparency statement to avoid ambiguity (pages 6-7).

“We originally hypothesized that reading about COVID-related acts of kindness would not only avoid the adverse impacts of exposure to COVID information, but actually have a positive effect on mood compared to the no information control condition. This hypothesis was not supported, perhaps because it is not possible for people to separate the positivity of kind acts from the negative context that inspired them. In order to test that hypothesis, we included an additional “amusement” condition, which was intended to rule out the explanation that the positive effects of the kindness condition were simply due to it increasing positive affect. For reasons of parsimony, and to avoid confusion, we do not report the responses from the amusement condition, or the responses on measures that are germane to this unsupported hypothesis (that the kindness condition would have more positive outcomes than the no information control condition, e.g., elevation). For more information, see the data and materials, which are posted online”.

 We have also updated the participants subsection to make it even clearer that we’re only reporting data from the kindness, covid and no-information condition, and not the amusement condition. 

We report the findings from the participants (N = 299) who completed the study in the three conditions relevant to our analyses (Mage = 29, SD = 12; 74% female, 17% male, 9% identified in another way; 86% White, 1% Black, 3% Asian, 6% Mixed, 4% Other); we do not report the results from a fourth (“amusement”) condition (see Transparency Statement above). 

e) The limitations section does not really name methodological limitations of the current study (e.g. relatively homogenous sample). I'd suggest expanding this section.

RESPONSE: In line with your suggestion, we now identify several key limitations of the present work. On pages 17-18 of the revised manuscript with now state: 

One strength of our Twitter study is that we utilised real Twitter feeds that continuously updated with new content, providing a naturalistic design that increases the external validity of our results. However, this design also means that we had no control over the content, and each participant may have experienced slightly different stimuli, even compared to other participants in the same condition. We do not know how objectively negative the COVID-information feed was, or how objectively positive the COVID-kindness feed was, and more work is needed to understand what features of content affect participants’ emotional responses. 

Notably while our research was not limited to samples comprised of undergraduate psychology students or MTurk participants, the current research is limited by its use of two relatively homogenous samples, which featured primarily young white and female participants, thereby narrowing the generalisability of the results (Study 1: Mage = 29, SD = 12; 74% white; Study 2: Mage = 26, SD = 9; 59% female). Indeed, some research has found that younger females have fared worse than their male counterparts in terms of mental health and stress during the pandemic [14, 36]. On the other hand, [37] found males experienced larger reductions in positive affect than females during the pandemic, and [14] found frequent social media use during the pandemic was associated with negative mental health consequences for both males and females. Taken together, it is unclear what effect a more balanced gender sample may have had on the results. With respect to homogeneity of age, given that older adults are able to regulate emotions more effectively than younger adults [38,39], we might speculate that an older sample may not have been as negatively impacted by brief exposure to COVID-info on social media. However, older adults may also be less likely to use social media [40], which reduces the importance of this limitation.

Once again, we thank you for the time and attention you have dedicated to our work. Please let us know if you require any further changes or information. 

Sincerely, 

Kathryn Buchanan, Lara Aknin, Shaaba Lotun, and Gillian Sandstrom

---

## [Decision Letter · Decision Letter 1]

9 Sep 2021

Brief exposure to social media during the COVID-19 pandemic: Doom-scrolling has negative emotional consequences, but kindness-scrolling does not

PONE-D-21-14745R1

Dear Dr. Buchanan,

We’re pleased to inform you that your manuscript has been judged scientifically suitable for publication and will be formally accepted for publication once it meets all outstanding technical requirements.

Kind regards,

Barbara Guidi

Academic Editor

PLOS ONE

Additional Editor Comments (optional):

Reviewers' comments:

Reviewer's Responses to Questions

**Comments to the Author**

1. If the authors have adequately addressed your comments raised in a previous round of review and you feel that this manuscript is now acceptable for publication, you may indicate that here to bypass the “Comments to the Author” section, enter your conflict of interest statement in the “Confidential to Editor” section, and submit your "Accept" recommendation.

Reviewer #1: All comments have been addressed

2. Is the manuscript technically sound, and do the data support the conclusions?

Reviewer #1: Yes

3. Has the statistical analysis been performed appropriately and rigorously? 

Reviewer #1: Yes

4. Have the authors made all data underlying the findings in their manuscript fully available?

Reviewer #1: Yes

5. Is the manuscript presented in an intelligible fashion and written in standard English?

Reviewer #1: Yes

6. Review Comments to the Author

Reviewer #1: (No Response)

7. PLOS authors have the option to publish the peer review history of their article (what does this mean?). If published, this will include your full peer review and any attached files.

Reviewer #1: No

---

## [Editor Report · Acceptance letter]

20 Sep 2021

PONE-D-21-14745R1 

Brief exposure to social media during the COVID-19 pandemic: Doom-scrolling has negative emotional consequences, but kindness-scrolling does not 

Dear Dr. Buchanan:

I'm pleased to inform you that your manuscript has been deemed suitable for publication in PLOS ONE. Congratulations! Your manuscript is now with our production department. 

Kind regards, 

on behalf of

Dr. Barbara Guidi 

Academic Editor

PLOS ONE